# Tea Tree Oil Induces Systemic Resistance against Fusarium wilt in Banana and Xanthomonas Infection in Tomato Plants

**DOI:** 10.3390/plants9091137

**Published:** 2020-09-02

**Authors:** Ronaldo J. D. Dalio, Heros J. Maximo, Rafaela Roma-Almeida, Janaína N. Barretta, Eric M. José, Agnelo J. Vitti, Daphna Blachinsky, Moshe Reuveni, Sérgio F. Pascholati

**Affiliations:** 1Departament of Plant Pathology and Nematology, University of São Paulo (USP/Esalq), Piracicaba 13400-900, Brazil; rdalio@ideelab.com.br (R.J.D.D.); molecularcrinkler@gmail.com (H.J.M.); rafaelacroma@gmail.com (R.R.-A.); janaina.barretta@usp.br (J.N.B.); 2STK Bio-Ag Technologies Ltd., Petah Tikva 4922297, Israel; ericm@stk-ag.com (E.M.J.); agnelo.vitti@gmail.com (A.J.V.); Daphna.Blachinsky@stk-ag.com (D.B.); moshe@stk-ag.com (M.R.); 3Shamir Research Institute, University of Haifa, Katzrin 12900, Israel

**Keywords:** tea tree oil (TTO), *Fusarium oxysporum*, *Xanthomonas* spp., resistance (ISR) and priming effect

## Abstract

The essential tea tree oil (TTO) derived from *Melaleuca alternifolia* plant is widely used as a biopesticide to protect crops from several plant-pathogens. Its activity raised queries regarding its ability to, not only act as a bio-fungicide or bio-bactericide, but also systemically inducing resistance in plants. This was examined by TTO application to banana plants challenged by *Fusarium oxysporum* f. sp. *cubense* (Foc, Race 1) causing Fusarium wilt and to tomato plants challenged by *Xanthomonas campestris*. Parameters to assess resistance induction included: disease development, enzymatic activity, defense genes expression correlated to systemic acquired resistance (SAR) and induced systemic resistance (ISR) and priming effect. Spraying TTO on field-grown banana plants infected with Foc and greenhouse tomato plants infected with *Xanthomonas campestris* led to resistance induction in both hosts. Several marker genes of salicylic acid, jasmonic acid and ethylene pathways were significantly up-regulated in parallel with symptoms reduction. For tomato plants, we have also recorded a priming effect following TTO treatment. In addition to fungicidal and bactericidal effect, TTO can be applied in more sustainable strategies to control diseases by enhancing the plants ability to defend themselves against pathogens and ultimately diminish chemical pesticides applications.

## 1. Introduction

Plants are exposed to various pathogens in their environment and have developed immune systems with multiple defense layers to prevent infections [1,2]. However, often pathogens overcome these resistance barriers, infect plants and cause disease. Pathogens that cause diseases on economically important crop plants incur huge losses to the agriculture industry and important ecosystems [3,4].

Chemical pesticides have often been used to control diseases in plants, but this conduct is associated to negative environmental impacts, potential human exposure to pesticides, and deposition of residues on the plant organs. However, the effectiveness of synthetic pesticides has been reduced by the frequent development of resistance by the pathogens. Hence there is a great demand for safer, alternative and effective agents [5].

Bananas, the world’s most important fruit in terms of production, volume and trade [6] and among the world’s top 10 staple foods, is seriously threatened by Fusarium wilt (FW) caused by the fungus *Fusarium oxysporum* f. sp. * cubense* (Foc). The fungus penetrates the plant through the root system and may causes plant death. After the death of the mother plant, the fungus continues infecting the daughter plant and remains active on the clump for years [7,8,9]. 

The tomato production is yearly affected by disease caused by *Xanthomonas spp.* [10]. The disease is characterized by necrotic lesions on the leaves, stems, petals, flowers, and fruit [11]. During the initial stages of symptom development, circular water-soaked lesions appear, which later dry and turn dark brown to black with a wet to greasy appearance [12].

Both microorganisms are important because of the losses they can cause in crop production. No chemicals are known to control Fusarium wilt [8] and they are very limited for bacterial diseases. 

An alternative procedure to protect plants against disease is to activate their inherent defense mechanisms by specific biotic or abiotic elicitors. Plants can be induced to switch on defense reactions to a broad range of pathogens as a result of prior exposure to pathogens or to various chemicals or physical stresses. Induced resistance is expressed locally, at the site of the infection or systemically, at sites located far from the initial infection. Upon recognition of the initial stimulus by the plant, a signal transduction pathway is set in motion, that includes intra and intercellular signals, and results in the activation of defense mechanisms, mostly by the expression of new genes [13,14].

During the interactions with microorganisms and molecular patterns from the environment, plants produce several hormones that act as signals that trigger the production of antimicrobial compounds and activate defense in general [15,16]. Upon recognition, plants produce salicylic acid, jasmonate and/or ethylene as major defense signals [17]. Salicylic acid (SA), for example, plays an important role in defense in biotrophic interactions. SA biosynthesis is induced in the infection site where it contributes to reactive oxygen species (ROS) production and cell death in a positive feedback [18,19,20,21]. Jasmonate (JA) and ethylene (ET), on the other hand, are generally associated to the regulation of defenses against herbivores and necrotrophic pathogens, although exceptions can be seen [22,23]. The activation of both jasmonic acid and ethylene pathways is frequently associated with induced systemic resistance (ISR) [20]. Thus, the genes for the three defense related pathways can be considered as molecular markers to study induced resistance in plants.

Frequently, SAR or ISR are accompanied by the priming effect. Priming is a phenomenon in which a plant activates their defense system by recognizing a molecular pattern from the environment in the absence of a pathogen and no development of disease, the defense system is deactivated and the plant turns back to homeostasis [14]. However, if the plant is subsequently challenged by a pathogen or abiotic stress, through epigenetics, it can activate their defense system more rapidly and robustly [24,25,26]. Inducing priming is preferable rather than the classic resistance induction where the plants may allocate too much energy on defense and that may affect growth and production. [24,25].

The global search for plant-protection solutions, that are both environmentally safe and efficacious, is an important aspect of sustainable agriculture. This is driven by the need to supply food to the ever-growing world population, and the call for chemical load reduction.

The essential tea tree oil (TTO) extracted from *Melaleuca alternifolia* plant, contains many components, mostly terpenes and their alcohols, was found to be an effective antiseptic, bactericide [27,28,29] and more recently as an effective fungicide [30,31,32]. Based on TTO, as an active ingredient, a natural fungicide Timorex Gold^®^ (22.3 EC W/V), was prepared in order to enable the use of TTO on plant tissue [33]. In numerous crops, including bananas and fruit trees, this product was found effective against a broad range of plant-pathogenic fungi [32,33,34,35]. The high effectiveness of TTO raised questions regarding its ability to perform as a resistance inducer. TTO is highly applied in banana and tomato plants worldwide, however very little is known about its mode of action in these hosts physiology.

Thus, this study was undertaken to examine the induction of systemic resistance in field-grown banana plants to Fusarium wilt and greenhouse-grown tomato plants to bacterial disease. Symptoms, peroxidase and β-1,3-glucanase activities, and the expression of different marker genes related to SAR and ISR were evaluated in response to early treatment with the TTO.

## 2. Results

### 2.1. Efficacy of TTO Compared to Chemical Fungicides in Field-Grown Banana Plants

In order to assess the efficacy of TTO to protect field banana against Foc we have tested it in comparison with triazole-based fungicides. In addition, we have treated the mother plants and analyzed disease symptoms progression in daughter plants to check for systemic effect. The incidence of *Fusarium* wilt-positive daughter banana plants obtained from triazole-treated (Nativo® tebuconazole and trifloxystrobin) mother plants resulted in 10 out 25 daughter plants (40%) being infected by Fusarium wilt. However, when mother banana plants were treated with TTO, only 2 out of 25 daughter plants (8%) were infected by Fusarium Wilt (Figure 1).

In a second experiment, evaluation of daughter banana plants obtained from treated mother plants one year after the first application of defensives, showed that treating mother banana plants with triazole-based fungicides resulted in 32 out 50 daughter plants (64%) infected by Fusarium wilt, while mother plants treated with TTO resulted in only 14 out of 50 daughter plants (28%) affected by Fusarium wilt. Moreover, in a second evaluation of the daughter plants, which was performed after two years from the first application of the mother plants, showed that 85% of daughter banana plants, obtained from triazole-based treatment were affected by Fusarium wilt, while only 12% of daughter plants originating from TTO-based treatment applied to mother plants were affected by the disease. 

### 2.2. Induction of Enzyme Activities in Banana Plants

Spraying of the plants with formulated TTO at 5, 10 and 15 days prior to inoculation with *Fusarium oxysporum* (Foc TR1) significantly increased, in a time-dependent manner, the enzymatic activity of guaiacol peroxidase and β-1,3-glucanase compared to untreated inoculated plants (Figure 2A,B). 

### 2.3. Expression of SAR and ISR Defense-Related Genes in Banana

The relative expression of genes related to the SAR (*NPR1, PR1, PR2*, *PR3* and *BSMT1*) and to the ISR pathways (*MYC2* and *ACS*) in leaves of banana plants were analyzed in non-treated asymptomatic plants (control), TTO treated plants, symptomatic plants and TTO-treated symptomatic plants. Treatment with TTO significantly induced the mRNA expression of *NPR1*, *PR1* and *BSMT1* in the pathways regulating salicylic acid, jasmonate and ethylene in healthy (asymptomatic) banana plants (Figure 3). In addition, the expression of four out of the five tested SAR-related genes were induced by the infection by *F. oxysporum* (INF). TTO treatment of symptomatic plants further increased the expression of *PR1, PR2* and *PR3* when compared to infected untreated (INF) plants (Figure 3). 

The TTO treatment of asymptomatic banana plants also resulted in induction of ISR-related genes, such as *MYC2, ACC* and *ERF1* (Figure 3). Infected plants with *F. oxysporum* (INF) showed prominent symptoms of Fusarium wilt, and some of the tested defense-related genes were upregulated during the infection, since plant tissues were destroyed by the fungal colonization and the plant might have recognized damage-associated molecular patterns (DAMPs) that, in turn, activated the defense reaction. However, TTO treatment of infected plants (TTO + INF) caused changes in the response pattern. The genes *PR1, PR2* and *PR3* that encode for pathogenesis-related proteins were highly overexpressed as a result of the TTO treatment in the TTO + INF treatment when compared to untreated INF plants (Figure 3).

### 2.4. Induction of Resistance in Tomatoes vs X. Campestris pv. Vesicatoria

Tomato plants were treated with TTO and inoculated with *Xanthomonas campestris.* Symptoms were recorded (Figure 4). Non-inoculated TTO treatment showed no symptoms and no signal of phytotoxicity. Inoculated plants showed symptoms and were visibly impaired by the infection when compared to control plants. The TTO treatment alleviated the symptoms caused by the bacteria.

A gene array analysis performed on leaf samples from tomato plants showed that TTO pre-treatment in asymptomatic (TTO) tomato plants led to the induction of the expression of genes related to the resistance pathways, such as R-genes (*HCR2* and *CCNBS*) and other genes related to the SAR pathway (*WRKY* transcription factor, *NPR1*, *PR1* and *PR2*) and the ISR pathways, (*JAZ1*, *LOX2, MYC2, ACC, ERF1* and *EIN3*) (Figure 5 and Figure 6). The overexpression of these resistance-mediating genes was even stronger in TTO-treated plants showing symptoms of bacterial disease at 3 days post inoculation with *X. campestris* (TTO + INF). Inoculated plants with *Xanthomonas spp.*, without treatment with TTO, did not show any significant overexpression of the above-mentioned genes at 3 dpi (Figure 5). In fact, the expression of *NPR1, ERF1* and *EIN* were downregulated because of the bacterial infection at this stage (Figure 5). The bacterial infection led to an up-regulation of several genes (mainly R-genes and ISR-associated genes) at 10 dpi (Figure 6). Yet, TTO pre-treatment of plants showing bacterial disease induced an even greater expression of genes, including overexpression of SAR-related genes (Figure 5 and Figure 6). This overexpression was also higher than the gene induction observed at 3 dpi in infected plants treated with TTO (Figure 5A). The effect of TTO in asymptomatic plants at 10 dpi was not as strong as at 3 dpi, indicating a negative feedback mechanism (Figure 5 and Figure 6). Furthermore, TTO treatment showed overexpression of genes related to the phenylpropanoid pathway.

### 2.5. TTO Primes Tomato Plants to Have a Defense Reaction to a Subsequent Challenge

A gene array analysis performed on leaf samples from tomato plants showed that a mechanical wound made in the plants led to an induction of R-genes (*HCR2* and *CCNBS*) and genes related to the ISR pathway (*JAZ1, LOX2, MYC2, ACS, ERF1* and *EIN3*) at 24 h post wounding (Figure 7). These results are indicative of a priming phenomenon. The TTO pre-treatment of the wounded plants lead to a strong overexpression of SAR-related genes (*WRKY* transcription factor, *NPR1, PR1* and *PR2*) in addition to the induction of R-genes and ISR-related genes (Figure 7).

## 3. Discussion

Tea tree oil (TTO) was approved by the European Union (EU) and included in the positive list of the EU, in Annex I of Directive 91/414/EEC for registration of Plant Protection Products. TTO is classified as a low risk substance in Europe, for which establishment of Maximum Residue Limits (MRL) is not required. As no residue has been established, no Protected Health Information (PHI) is required. However, it depends on local authorities’ regulation (currently ranged between 0–4 days). In addition, we decided to choose TTO because it is extracted by steam distillation from renewable plants easy to grow, cost, efficacy, environmental and human consideration. This oil is in large use in cosmetics and in medicine.

The essential tea tree oil (TTO) was shown to have antiseptic, fungicide and bactericide properties [27,28,29,36] and in the past decade has been used as an effective fungicide on numerous crops against plant pathogens [30,31,33,37]. The fungicidal and antimicrobial activities of TTO against fungal and other pathogens arise from its ability to alter the permeability of membrane structures [27,29,36,38]. In yeast cells and isolated mitochondria the extract components of the tea tree (*M. alternifolia*) destroy cellular integrity, inhibit respiration and ion transport processes, and increases membrane permeability [27,29,36,38]. Previous studies showed that formulated TTO (Timorex Gold^®^) effectively controlled black Sigatoka in banana plantations [33,34,37] by exhibiting a strong curative activity based upon disruption of cell membrane and destruction of the wall of the hyphae of *Mycosphaerella fijiensis* in infected leaves [37]. Nevertheless, no reports exist in the literature showing that TTO can be an inducer of systemic resistance in plants against pathogens.

Thus, we report here that TTO efficiently protected systemically banana plants against Fusarium wilt caused by the soil-born pathogen *F. oxysporum* f. sp. *cubense*. It was effective when applied to the foliage of infected mother plants and systemically protected the daughter plants as seen in the following year (Figure 1). The results demonstrated that spraying of TTO on foliage of mother plants infected or exposed to Foc significantly reduced the incidence of Fusarium wilt in daughter banana plants. Since effective biological, chemical and cultural measures against Fusarium wilt are not available or are very limited [8], the induced systemic resistance by TTO seems to be a unique and desirable phenomenon to banana growers. This is mainly because the translocation of TTO by itself from the mother to daughter plants does not seem so feasible and has not been shown in other plants. Therefore, how does TTO acts to induce systemic resistance?

Elicitors may trigger defense reactions by mimicking interactions of natural microbe molecular patterns or defense signaling molecules with their respective cognate plant receptors or by interfering with other defense signaling components. Often the term “plant activators” is used for molecules that can protect plants from diseases by inducing immune responses. Prior applications of TTO to young banana plants induced the activities of the PR proteins (Figure 2A,B) guaiacol peroxidase (Figure 2A) and β-1,3-glucanase (Figure 2B). Since these PR-proteins are related to the defense mechanisms of plants against pathogens, these results indicate that TTO acts as an inducer of such defense mechanisms in the treated plants. This suggests that PR-proteins may contribute to the resistance of banana against Fusarium wilt. Because TTO did induce the activities of PR-proteins in banana we suggest that it operates via the SA-pathway (Figure 2 and Figure 3B).

The TTO significantly induced the mRNA expression of *NPR1, PR1* and *BSMT1* in healthy (asymptomatic) banana plants (Figure 3A) and further increased the expression of *PR1, PR2* and *PR3* in symptomatic plants when compared to infected untreated (INF) plants. In asymptomatic banana plants, TTO also induced ISR-related genes, such as *MYC2, ACS* and *ERF1* (Figure 3A,B).

Similar results were obtained when TTO was applied to tomato plants. Treatment of asymptomatic (TTO) plants led to the induction of the expression of genes related to resistance pathways, such as R-genes (*HCR2* and *CCNBS*) and other genes related to the SAR and ISR pathways (Figure 5 and Figure 6). The overexpression of these resistance-mediating genes was even stronger in TTO-treated plants showing symptoms of bacterial spot 3 days after *X. campestris* inoculation. Inoculated plants with *X. campestris*, without treatment with TTO, did not show any significant overexpression of the above-mentioned genes at 3 dpi (TTO + 3dpi) (Figure 5). The bacterial infection led to an up-regulation of several genes (mainly R-genes and ISR-associated genes) at 10 dpi. This is probably because that at this time point, the infection was robust enough for the plant to recognize the pathogen and the bacterial-induced damage to the plant. Yet, TTO pre-treatment of plants showing bacterial disease was even higher, including overexpression of SAR-related genes. The effect of TTO in asymptomatic plants at 10 dpi was not as strong as at 3 dpi, indicating a negative feedback mechanism. Previous studies using the SAR inducers acibenzolar-S-methyl and harpin, applied each in combination with a bacteriophage, significantly reduced bacterial spot too [10].

These results demonstrate that TTO is an efficient resistance inducer, since it enhances the expression of marker genes in banana and tomato plants for both SAR and ISR pathways, via the three main defense related pathways, namely salicylic acid, jasmonic acid and ethylene-mediated pathways (Figure 5B and Figure 6B). 

The first crucial step for a plant is to recognize that it is been attacked [39]. Priming is a strategy used to improve the defensive capacity of plants by activating the plant defense mechanisms prior to the challenge stress. Such activation may include changes at the physiological, transcriptional, metabolic and/or epigenetic levels. Thus, upon facing a subsequent challenge, the plant effectively mounts a faster and/or stronger defense response that results in increased resistance and/or stress tolerance [40,41,42]. Our data also showed that TTO pre-treatment of the wounded tomato plants led to a strong overexpression of SAR-related genes (*WRKY* transcription factor, *NPR1, PR1* and *PR2*) in addition to the induction of R-genes and ISR-related genes (Figure 7). However, the activation of the priming in the tested plants is still unknown in specific to cell receptors for TTO. In this sense, it addresses the biology of innovative systems, such as genomics, proteomics and phenomics currently recognized as essential tools to understand the molecular mechanisms underlying plant responses to environmental changes. There are a few published studies on understanding the interaction between plant receptors and priming activation [43]. Thus, TTO can also serve as potent tool or probe in basic research approaches for expanding our knowledge in plant immunity.

## 4. Materials and Methods

### 4.1. Banana vs F. oxysporum

For the banana vs. *F. oxysporum* interaction, we have studied the effect of TTO in three different field experiments, targeting induction of systemic resistance. In the experiments 1, 2 and 3 the efficacy of TTO was compared to triazole fungicides against Fusarium wilt and evaluated in field-grown banana daughter plants following treatments with TTO or fungicides. We have recorded symptoms and disease incidence. In a further experiment, we have treated plants with TTO and studied SAR and ISR defense-related gene expression and enzyme activity. While the gene expression analyses were performed in plants harvested from field conditions, the enzymatic activity were rather studied in a more controlled conditions in greenhouse.

Tea tree oil was used in all trials as an emulsifiable concentrated formulation (Timorex Gold^®^, 22.3 EC W/V; STK Bio-ag Technologies, Petah Tikva, Israel). The following fungicides were tested for comparison in banana: Nativo^®^ 75WG (Bayer CropScience Ltd., Monheim, Germany), a premixed fungicide containing tebuconazole and trifloxystrobin, and the sterol inhibitor fungicides Sico^®^ 250CE (Syngenta, Basel, Switzerland), containing difenoconazole, or Tilt^®^ 250 EC (Syngenta, Basel, Switzerland) containing propiconazole.

### 4.2. Induction of Systemic Resistance in Field-Grown Banana Plants

The first experiment was conducted in Brazil in Delfinopolis, Minas Gerais on banana plants cv. Prata Gorotuba affected by Fusarium Wilt, caused by *F. oxysporum*. Plants were treated with formulated TTO (Timorex Gold) applied at 0.6 L/ha or Nativo^®^ applied at 0.5 L/ha. Treatments included four foliar applications of each of the materials to the leaves, at monthly intervals in areas of 36 ha for each treatment. Twenty five banana families, with the mother plant expressing clear symptoms of wilt and healthy daughter plants with 4 to 6 leaves stage were randomly marked in each treated plot. Evaluation of Fusarium wilt-positive or negative daughter plants was performed visually, at one year after the first application of treatments and by an incision in the trunk of the daughter plants.

The second experiment and its repetition were conducted in Brazil in Jacupiranga, São Paulo on banana plants cv. Prata Litoral, affected by Fusarium wilt (Foc TR1). Plants were treated with the triazole-based fungicide Tilt (propiconazole) at 450 mL/ha and formulated TTO at 0.6 L/ha. For both fungicides and TTO application, we have also added the biostimulants Biozyme® and Kfol®. The biostimulants serve to increase the amount and quality of fruit produce rather than treating plant infections. The treatments involved five foliar applications of the fungicides by spraying at 1.5–2.0 months intervals. Each treated area was composed by 7.6 ha in which 50 banana families, with the mother plant expressing clear symptoms of wilt and healthy daughter plants with 4 to 6 leaves stage were randomly marked. Evaluation of the incidence of daughter banana plants affected by Fusarium wilt was made one and two years after the first application of the treatments, as described above. The symptoms were clearly visible after performing a cross section in daughter plants. Dark spots and necrosis were visible in tissues.

### 4.3. Analysis of PR Proteins Activity in Banana Plants 

Banana plants cv. Prata were grown in a greenhouse and starting at third leaf stage, they were sprayed with formulated TTO until run-off at 5, 10 or 15 days prior to inoculation with Foc tropical race 1. Conidial suspension containing 1 × 10^6^ spores/mL of Foc was used for inoculation. We have wounded the plants roots with a shovel, and applied 200 mL the spore suspension in the wounded area. Thirty days after inoculation, roots were removed from each of the ten plants and used for analysis.

### 4.4. Activities PR Proteins in Banana Root Samples

Separated roots were macerated together and submerged in liquid nitrogen until the formation of powder. Then sodium phosphate buffer (100mM, pH 7.5) was added and the samples were centrifuged at 20,000 *g* for 30 min at 4 °C. The supernatant was collected and used to perform the biochemical analyses described below, while keeping in ice. The activity of guaiacol peroxidase was determined in a reaction mixture consisting of 2.9 mL of the reaction buffer (10 mM sodium phosphate buffer at pH 7.5, 2.3 mM guaiacol and 2.9 mM H_2_O_2_) and 0.1 mL of the plant extract. The conversion of guaiacol to tetra-guaiacol by guaiacol peroxidase was monitored spectrophotometrically at 470 nm during 1 min, at 15 s intervals. The activity of β-1,3-Glucanase was determined by the quantification of glucose released from laminarin as a substrate. Total of 150 μL of the plant extract was mixed with 150 μL of 0.2% laminarin dissolved in 10 mM sodium phosphate buffer (pH 7.5). The reaction mixture was incubated at 37 °C for 3 h. One and half mL of *p*-hydroxybenzoic acid hydrazide (PAHBAH) was added to the incubated reaction mixtures. The reaction was terminated by heating at 100 °C for 10 min. A negative control sample was prepared by incubating the plant extract for 3 h (without laminarin) followed by addition of laminarin immediately before the termination of the reaction. Glucose concentration was determined spectrophotometrically at 410 nm. The results obtained in the negative control were subtracted from the results of the tested samples in order to determine the glucose content released from laminarin independently from basal levels of glucose in the root samples, namely to determine the enzymatic activity of β-1,3-glucanase.

### 4.5. Expression of SAR and ISR Defense-Related Genes in Banana 

Full grown banana plants in Delfinópolis-MG, Brasil, were treated three times by foliar application with formulated TTO at 44 ml/L solution. Twelve days after the last application, the third fully-expanded leaf from each daughter plants was harvested for RNA extraction and gene expression analyses. Four treatments were harvested such as: control plants, not treated with TTO and showing no symptoms (Con); TTO treated plants and showing no symptoms (TTO); plants showing symptoms and not treated with TTO (INF); plants showing symptoms and treated with TTO (TTO + INF). After sampling, leaves were immediately frozen in liquid nitrogen until further analysis.

### 4.6. Induction of Resistance in Tomatoes vs X. campestris pv. vesicatoria

For the tomato vs *X. campestris* pv. *vesicatoria* interaction, we have performed greenhouse experiments in which we recorded symptoms and analyzed activation of SAR and ISR defense related genes and the priming effect.

#### 4.6.1. Plants

Seeds of Top Seed Italian tomato variety were pre-germinated in autoclaved Petri dishes containing filter paper moistened with sterile distilled water. The dishes with seeds were kept in bio-oxygen demand (B.O.D.) incubator at 25 °C and photoperiod of 12 h for 4 days. The germinated seeds were transferred to plastic pots containing substrate for planting and kept in a greenhouse for development until used for inoculation at 28 days old stage.

#### 4.6.2. Pathogen and Inoculation

A culture of *X. campestris* was grown in nutrient-agar (NA) medium, which consists of: 2 g yeast extract; 5 g peptone; 5 g sodium chloride and 15 g agar and kept in a B.O.D. incubator at 28 °C in the dark for about 24 hours, to allow the bacterium reach a high rate of multiplication. 

Plants were kept in a humid chamber for 24 h prior to inoculation in order to favor stomata opening and, consequently, pathogen penetration. The bacterial suspension was obtained by scraping the culture medium on the Petri dish with a flanged Drigalski handle, and sterile distilled water containing 0.85% NaCl (saline solution: to maintain cell integrity and viability). The suspension obtained was kept under stirring until complete dissolution of the colonies in the saline solution. Inoculation was performed by spraying all leaves until run-off. Plants were then kept in a humid chamber for additional 48 hours.

#### 4.6.3. Experimental Design

Plants with five true leaves either inoculated with *Xanthomonas campestris* or not inoculated and treated with formulated TTO at 0.5% (*v/v*) in water, applied once or not treated were used. The youngest fully expanded leaf was sampled three and ten days post inoculation (dpi) from each of the following four treatments: (A) control plants (CON); (B) TTO-treated plants (TTO); (C) *X. campestris* inoculated plants (INF); and (D) plants treated with TTO and inoculated with *X. campestris* performed 72 hour after TTO application (TTO + INF).

To test the priming effect of TTO, tomato plants were treated with formulated TTO and after mechanically wounding the plants in order to simulate a pathogen attack. The experiment was divided into four groups: (1) untreated and unwounded plants (Con); (2) plants treated with TTO at 0.5% (*v/v*) in water, applied once, and unwounded (TTO); (3) untreated plants mechanically wounded by stapling the youngest fully expanded leaf of the plant one time, with a stapler in the absent of staples (WOU); and (4) plants pre-treated with TTO as indicated above and mechanically wounded 72 h after TTO application (TTO + WOU). Leaf samples were taken 24 h post wounding (hpw).

### 4.7. RNA Extraction and Reverse Transcription

#### 4.7.1. RNA Extraction and Reverse Transcription of Banana Plants

The sampled banana leaf tissue was initially macerated in liquid nitrogen with the aid of pre-autoclaved mortar and pistil and frozen until the sample was transformed into a fine powder. About 500 mg of macerated plant tissue was centrifuged, re-suspended in 1500 µL extraction buffer (150 mM Tris-base, 4% SDS; 100 mM EDTA, 2% β-mercaptoethanol and 3% polyvinylpyrrolidone at pH 7.5 adjusted with saturated boric acid solution), vortexed and kept for 10 min in a water bath at 65 °C. The samples were then carefully stirred by inversion and allowed to cool to room temperature. The contents of the tube were approximately divided into two separate tubes and 66 µL potassium acetate (5 mM) and 150 µL absolute ethanol were added to each tube. Tubes were vortexed for 1 minute and then 850 µL chloroform: isoamyl alcohol (49:1; *v/v*) was added and the samples were vortexed again for 10 seconds. The samples were centrifuged at 12,000 × *g* for 20 min at room temperature. The supernatant was recovered into a new tube and 850 µL phenol: chloroform: isoamyl alcohol (25:24:1; *v/v/v*) was added. The samples were vortexed for 10 seconds and centrifuged at 12,000 × *g* for 15 min at room temperature. The supernatant was recovered into a new tube and 850 µL chloroform: isoamyl alcohol were added, vortexed for 10 s and centrifuged at 12,000 × *g* for 15 min at 4 °C. Again, the supernatant was recovered into a new tube and lithium chloride solution was added to a final concentration of 3 M. After gently swirling by inversion, the material was stored at -20 °C overnight. The samples were then centrifuged at 12,000 × *g* for 20 min at 4 °C and the supernatant was discarded. The pellets were washed with 500 µL of 70% ethanol twice and centrifuged at 12,000 × *g* for 10 min at 4 °C after each wash. The samples were resuspended in 10 µL diethylpyrocarbonate (DEPC)-treated water and stored in the ultra-freezer. RNA was spectrophotometrically quantified using Nanodrop spectrophotometer (Thermo Scientific, Waltham, MA, USA) [44,45].

#### 4.7.2. RNA Extraction and Reverse Transcription of Tomato Plants

RNA purification of tomato plants was carried out using the MasterPure Plant RNA Purification kit (Epicentre Biotechnologies, Madison, WI, USA), followed by DNaseI (Thermo Fisher Scientific, Waltham, MA, USA) treatment, according to manufacturer’s recommendations. Total mRNA was reverse transcribed by using oligo-dT primers in a 20 µL reaction volume by using RevertAid First Strand cDNA Synthesis Kit (Thermo Fisher Scientific) and 1 µg total DNA-free RNA. cDNA was diluted (1:20), and 1 µL of the diluted cDNA was used in a 13 µL reaction volume containing 6.75 µL of Go-Taq qPCR Master mix, and 0.75 µM (1 µL) of each primer [44,46].

### 4.8. Gene Expression Analysis

Quantitative real time PCR (qPCR) was performed in the Applied Biosystem 7500, in triplicates under the following conditions: 95 °C for 20 s and 40 cycles of 95 °C for 3 s and 60 °C for 30 s. Specificity of the primers used to quantify the expression of each gene was confirmed by performing melt curve analysis, in which the temperature of the sample was gradually raised from 65 °C to 95 °C in 0.5 °C steps for 5 s each [44,46]. The analyzed genes related to plant’s defense pathways, such as salicylic acid (SA), ethylene (ET) and jasmonic acid (JA), normal physiological conditions, and other general defense-related genes. *Arabidopsis thaliana* gene sequences were obtained from TAIR (http://www.arabidopsis.org) and used for blast search of hosts (e.g., banana, tomato) orthologous genes by BLASTX on Phytozome (http://www.phytozome.net). The sequences with the highest hit score based on identity and query cover, i.e., lowest E-value were selected for analysis. Primers were designed using primer3 Plus software (http://www.bioinformatics.nl/cgi-bin/primer3plus/primer3plus.cgi), considering the following parameters: 40–60% GC content, 18–24 nucleotides in length, annealing temperature of 60 ± 2 °C, and 70–200 base pairs of amplicon length (Table 1).

### 4.9. Analysis of Housekeeping Reference Genes

The following housekeeping genes were used for the normalization of gene expression in plants: Glyceraldehyde-3-phosphate dehydrogenase C2 (GAPC2), NADP^+^-dependent isocitrate dehydrogenase (NADP-IDH), homolog to DIM1 (YLS8), cyclophilin (CYP) and F-box family protein (FBOX) [47] (Table 1).

### 4.10. Gene Array

RNA samples were reverse-transcribed to cDNA by using routine methods. The cDNA samples were diluted 1:10 prior to use in the qRT-PCR reaction. Each diluted cDNA sample (1 μL) was pipetted three times in a 96-well plate suitable for real-time PCR readings, generating three technical repetitions for each sample. As a technical control, 0.1% diethylpyrocarbonate (DEPC)-treated water was used in three technical repetitions. Each gene of interest was studied in a different 96-well plate.

### 4.11. Statistical Analysis

For statistical analysis of gene expression, Cq and primer efficiency values were calculated from raw fluorescent data (Rn values) by using the Miner program Real-time PCR (http://ewindup.info/miner/) [44,48,49]. Housekeeping genes such as *CYP*, *FBOX* and *GAPC2* were used for normalization in plants [47]. Relative quantification (Rq) was calculated by using 2-ΔΔCq, and the significance of the results were tested by using the Kruskal–Wallis test [50]. All significant data regarding Rq was expressed in the figures in Log2. In this way, positive data would mean up-regulated genes and negative data would mean down-regulated genes. The different shades of red and green match the levels of expression of each gene.

## 5. Conclusions

Evidence shown in this manuscript indicates that TTO is able to prime tomato plants to have a strong defense reaction to subsequent challenges, such as mechanical wounding. These results also demonstrate that TTO provides protection to the plant independently of the fungicide effect.

Data in this paper also show that TTO can be consider as an efficient resistance inducer, since it has enhanced the expression of marker genes in non-symptomatic banana plants for both SAR and ISR at the three main defense related pathways—salicylic acid, jasmonic acid and ethylene. To the best of our knowledge, this is the first report on the effect of TTO as a resistance inducer, which opens new possibilities for this product to be used in strategies to control pathogens by decreasing the number of traditional defensive applications and inhibiting the spread of both *Fusarium oxysporum* in banana and *Xanthomonas campestris* in tomato as well as other type of pathogens in different crops.

## Figures and Tables

**Figure 1 plants-09-01137-f001:**
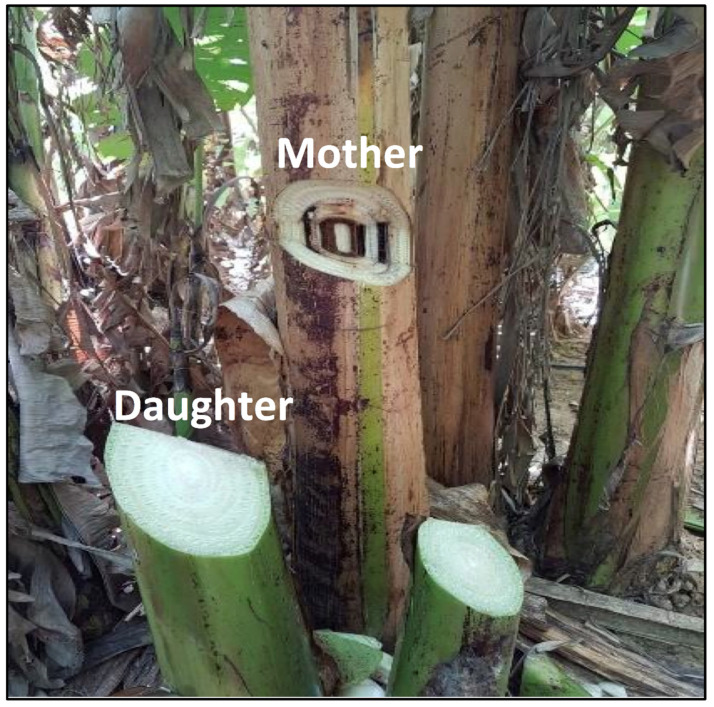
Acquired systemic resistance (SAR) and induced systemic resistance (ISR) via priming is activated by TTO in banana trees infected *Fusarium oxysporum f. sp. cubense* (Foc) race 1. Leaf sprays of TTO on infected mother banana trees induced protection against *Fusarium oxysporum f. sp. cubense* (Foc) race 1 in new plants (daughters) developed.

**Figure 2 plants-09-01137-f002:**
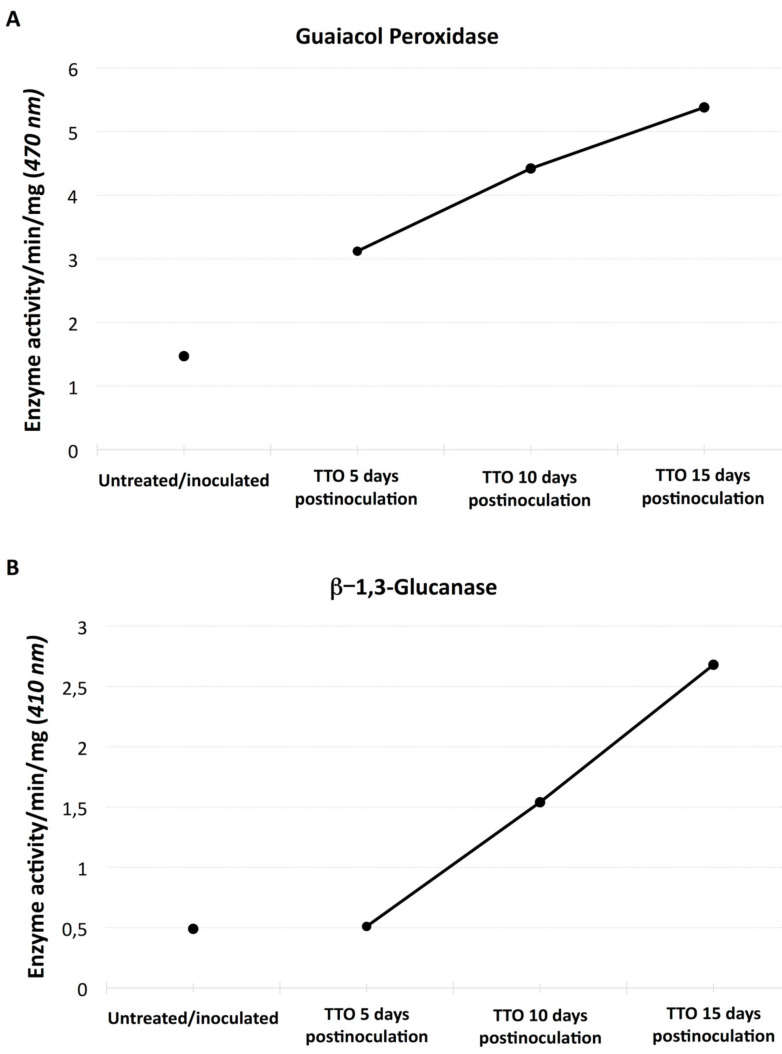
Effect of foliar applications of TTO on young banana plants in the activities of guaiacol-peroxidase (**A**) and B-1,3-glucanase (**B**).

**Figure 3 plants-09-01137-f003:**
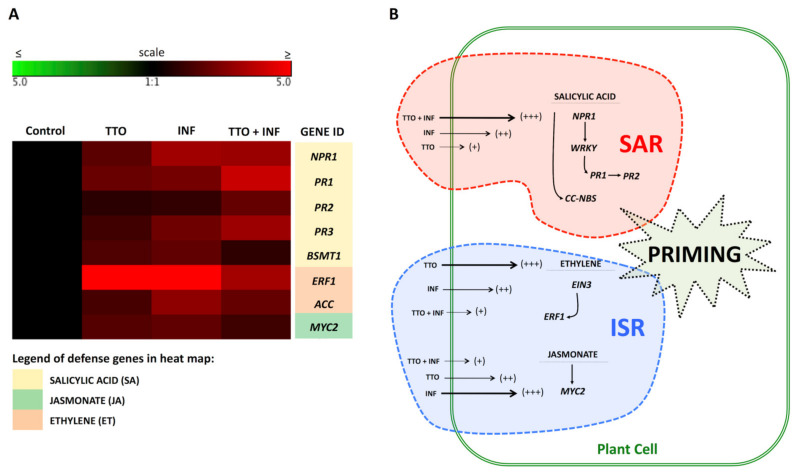
TTO induces systemic resistance to *Fusarium oxysporum* f. sp. *cubense* (Foc) in banana plants via SAR and ISR pathways. (**A**)—Gene expression in banana plants infected with Foc. The heat map illustrates the doubled changes in gene expression (log2 scale) in samples collected 15 days after the application of TTO. Different shades represent induced or repressed gene expression. Biological and technical triplicates were used in the analysis. Gene acronyms are listed in the right column. The defense routes are indicated by colors according to the legend. (**B**)—Scheme illustrating regulation of salicylic acid, ethylene and jasmonate pathways in the plant cell. Arrow sizes and scores (+++; ++; +) indicate the intensity of expression of genes related to SAR and ISR responses.

**Figure 4 plants-09-01137-f004:**
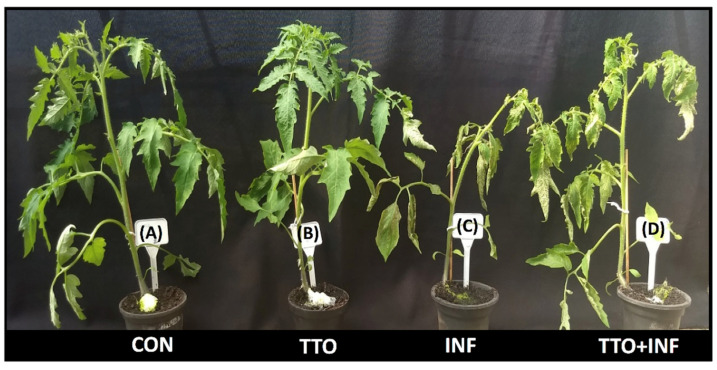
Effect of TTO treatment of tomato plants inoculated and non-inoculated with *X. campestris.* (**A**) control plants (CON); (**B**) TTO-treated plants (TTO); (**C**) *X. campestris* infected plants (INF); and (**D**) plants treated TTO and infected with *X. campestris* (TTO + INF).

**Figure 5 plants-09-01137-f005:**
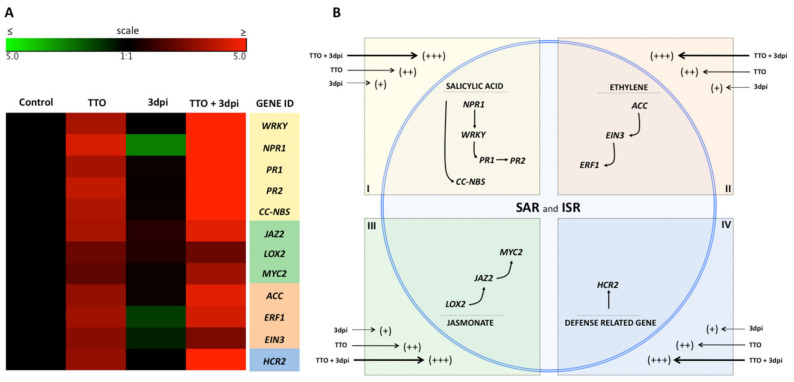
TTO induces systemic resistance to bacterial spot in tomato plants via SAR and ISR pathways. (**A**)—Gene expression in tomato plants infected by *Xanthomonas campestris*. The heat map illustrates the doubled changes in gene expression (log2 scale) in samples collected 3 days after application of TTO (TTO); 3 days postinoculation *X. campestris* (3 dpi); and 3 days after application of TTO more postinoculation *X. campestris* (TTO + 3dpi). Different shades represent induced or repressed gene expression. Biological and technical triplicates were used in the analysis. Gene acronyms are listed in the right column. The defense routes are indicated by colors according to the legend. (**B**)—Scheme illustrating regulation of salicylic acid, ethylene and jasmonate pathways in the plant cell. Arrow sizes and scores (+++; ++; +) indicate the intensity of expression of genes related to SAR and ISR responses.

**Figure 6 plants-09-01137-f006:**
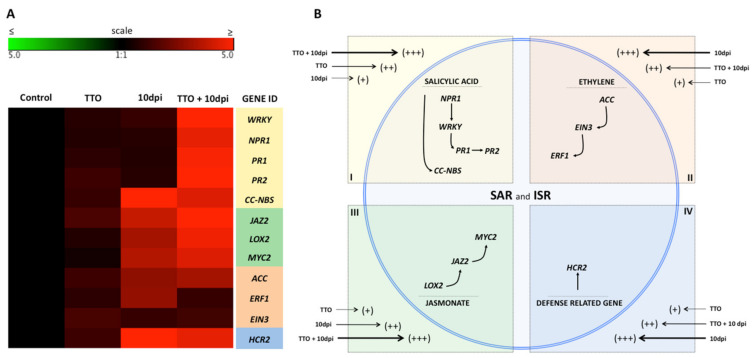
TTO induces systemic resistance to bacterial spot in tomato plants via SAR and ISR pathways. (**A**)—Gene expression in tomato plants infected by *X. campestris*. The heat map illustrates the doubled changes in gene expression (log2 scale) in samples collected 10 days after application of TTO (TTO); 10 days postinoculation *X. campestris* (10dpi); and 10 days after application of TTO more postinoculation *X. campestris* (TTO + 10dpi). Different shades represent induced or repressed gene expression. Biological and technical triplicates were used in the analysis. Gene acronyms are listed in the right column. The defense routes are indicated by colors according to the legend. (**B**)—Scheme illustrating regulation of salicylic acid, ethylene and jasmonate in the plant cell. Arrow sizes and scores (+++; ++; +) indicate the intensity of expression of genes related to SAR and ISR responses.

**Figure 7 plants-09-01137-f007:**
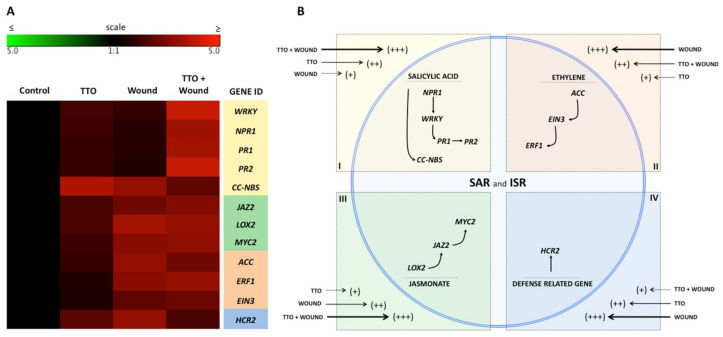
TTO induces systemic resistance in wounded tomato plants. (**A**)—The heat map illustrates the doubled changes in gene expression (log2 scale) in samples collected 10 days after application of TTO (TTO); 10 days post wounded (Wound); and 10 days after application of TTO more wounded (TTO + Wound). Different shades represent induced or repressed gene expression. Biological and technical triplicates were used in the analysis. Gene acronyms are listed in the right column. The defense routes are indicated by colors according to the legend. (**B**)—Scheme illustrating regulation of salicylic acid, ethylene and jasmonate pathways in the plant cell. Arrow sizes and scores (+++; ++; +) indicate the intensity of expression of genes related to SAR and ISR responses.

**Table 1 plants-09-01137-t001:** Lists of primers used in gene expression analysis.

***Lycopersicon esculentum mill***			
**ID_GENE**	**primerF**	**primerR**	**amplicon**
NPR-1	GGTCAGTGTGCTCGCCTATT	CACAGCTGGCCTACAAGCTAC	109
PR-1	GGATCGGACAACGTCCTTAC	GCAACATCAAAAGGGAAATAAT	124
PR-2	TATAGCCGTTGGAAACGAAG	TGATACTTTGGCCTCTGGTC	95
PR-3	CAATTCGTTTCCAGGTTTTG	ACTTTCCGCTGCAGTATTTG	88
BSMT1	GTTTAACGAGGCCGTTGATG	TCGTCAAGGAAACTGTCACG	200
ERF1	GGGGTCCTTGGTCTCTACTCA	GTAGCTTTTAAAACAGCAGCTGGA	112
ACC	AGCTACGTCAATGGCAGCAC	AGGAAGGGTGGGGACTTCTG	79
MYC2	ACCACATGAAAACAAAGCTGGAC	TCTCCGCCTCTACGTGGTTT	95
WRKY	ATCCTCGCCAGCAGTTAGCA	TCGTGGAGCTTTGCAAGGTAG	145
CC-NBS	TTCTGCAGAGTGTTCAATGGCAGC	CACAAACCCTTCAGCAACCCACAA	185
JAZ2	CCCCACCACCACTCAGACTAA	TATGGCGCTCTAGCCGTGT	117
LOX2	ACTGGTAGACCACCAACACGA	ACGCTCGTCTCTCGGTACAT	93
EIN3	CAGAAGTTCGACTAGAAACGGCTAT	TCCTCTGCTCTCAAGGATACAACA	147
HCR2	GCATGCAAGGACTGGTATGGA	TCTCGAGAAAAGGGAGGGATGA	124
***Musa paradisiaca***			
**ID_GENE**	**primerF**	**primerR**	**amplicon**
NPR-1	GGAGATCCACAAGTAGGTGAAGC	AGTCTTGCCAGAGCAACTCG	104
PR-1	TCCGGCCTTATTTCACATTC	GCCATCTTCATCATCTGCAA	126
PR-2	TCGCTGGGCTGTGGTAAGT	TCGCTGGGCTGTGGTAAGT	82
PR-3	GTCACCACCAACATCATCAA	CCAGCAAGTCGCAGTACCTC	150
BSMT1	GTTTAACGAGGCCGTTGATG	TCGTCAAGGAAACTGTCACG	200
ERF1	CCCAAATGTTGGTCCGTTTC	TCGCTGTCTTCCACGATTCA	79
ACC	GATGCTGCACATCGGCTAGT	GCCACCTGAATACGGCAGAC	123
MYC2	CGGATCTACCGACGTGGTCT	AGCGTCCGGAGAGCTAAAGT	82
**housekeeping genes**			
**ID_GENE**	**primerF**	**primerR**	**amplicon**
GAPC2	TCTTGCCTGCTTTGAATGGA	TGTGAGGTCAACCACTGCGACAT	80
DIM1	CGAAACCTGTATGCAGATGG	ACGGTTGAGGGATCGTAAAG	138
CYP	CGGATCTCAGTTCTTCGTCTG	ACTTTCTCGATGGCCTTGAC	111
FBOX	TTGGAAACTCTTTCGCCACT	CAGCAACAAAATACCCGTCT	112

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
