# Peer review of "Tea Tree Oil Induces Systemic Resistance against Fusarium wilt in Banana and Xanthomonas Infection in Tomato Plants"

_plants, 2020, doi:10.3390/plants9091137_

Round 1
Reviewer 1 Report
The article is interesting and has novelty but it needs a little more explanation on the way the research was done.
The results presented for the experiments conducted to demonstrate the efficacy of TTO in the control of Fusarium wilt of banana needs to be improve. First, you conducted only two experiments, not three. What you call experiment 3 is a repeated measure of experiment 2. The way the results are presented in Table 1 is confusing since for experiment 2 you applied 2 different triazoles What was the result for each one? Hoiw was the severity (%) determined? each mother plant should have more than one daughter plant, Which was your criteria to call a healthy or a diseased plant? Was there a statistical analysis performed?
In Mat & Met you indictaed that the bacteria used on tomatoes was Xanthomonas campestris. Was it pv. tomato? Why do you use Xanthomonas spp. in the text?
You need to describe how you did the evaluation of bacterial spot of tomato if you didi at all. It seems that the experiment was performed only to test for resistance response. No data on the severity of the disease obtained is presented in the text. You need to be more clear on this
Literature listed with numbers 23 and 41 are not cited in the text. Please check.
Corrections and other comments are in the document attached

Author Response
Dear Editors and Reviewers:
We are very happy with the positive reception of our manuscript, and we greatly appreciate the detailed, constructive critiques. Below we describe our response to each of the suggestions (designated by “>”). We also indicated the changes in the text and information inserted. We hope that you will find this version to be substantially improved and ultimately acceptable for publication.
Kind Regards,
Reviewers input and our responses:
Reviewer 1: The article is interesting and has novelty but it needs a little more explanation on the way the research was done.
>: We thank the reviewer 1 for all the questioning and suggestions below. We also appreciated the corrections and suggestions performed directly in the text. Everything was accepted and we believe it had impacted in a better quality of our manuscript.
The results presented for the experiments conducted to demonstrate the efficacy of TTO in the control of Fusarium wilt of banana needs to be improve. First, you conducted only two experiments, not three. What you call experiment 3 is a repeated measure of experiment 2. The way the results are presented in Table 1 is confusing since for experiment 2 you applied 2 different triazoles What was the result for each one? Hoiw was the severity (%) determined? each mother plant should have more than one daughter plant, Which was your criteria to call a healthy or a diseased plant? Was there a statistical analysis performed?
>: Indeed the ways we have presented these results were unclear. We have made some changes in order to clarify it in the results and in the MM sections. The Table was confusing, so we decided to take it out and explain all results in the text. We agree that the experiment 3 is just a second measurement of experiment 2, we have corrected as suggested. The severity was determined by analyzing symptoms in daughter plants, after cutting a section of it (As in Fig. 1). We have explained in detail in MM. We have selected the daughters at the 4 to 6 leaves stage. The diseased plants normally showed dark spots and necrosis in the tissues, easily recognizable. We have also explained that in the MM. No statistical analyses were performed.
In Mat & Met you indictaed that the bacteria used on tomatoes was Xanthomonas campestris. Was it pv. tomato? Why do you use Xanthomonas spp. in the text?
>: Is the X. campestris pv. vesicatoria. we have corrected that and removed all “Xanthomonas spp.” From the text. Inserted line: 199, 459, 461.
You need to describe how you did the evaluation of bacterial spot of tomato if you didi at all. It seems that the experiment was performed only to test for resistance response. No data on the severity of the disease obtained is presented in the text. You need to be more clear on this
>: We haven’t performed any severity measurements. We only check for resistance and symptoms development. We have removed a section on MM about severity.
Literature listed with numbers 23 and 41 are not cited in the text. Please check.
>: Thank you. We have corrected that. Inserted line: 89 and 354.
Corrections and other comments are in the document attached
>: Thank you very much!...we leave all changes highlighted in the text.
Reviewer 2 Report
The paper "Tea tree oil induces systemic resistance against
2 Fusarium wilt in banana and Xanthomonas
3 infection in tomato plants" is an interesting, well-written work and is the first report on the effect of TTO as a resistance inducer. This work opens, probably, as hypothesized by the authors, new possibilities for this oil to be used in strategies to control pathogens by decreasing the number of
traditional defensive applications.
The only advice I would like to give to the authors is to mention literature data in the discussions that demonstrate how safe this tea tree oil is from both an environmental and human health point of view.
Author Response
Dear Editors and Reviewers:
We are very happy with the positive reception of our manuscript, and we greatly appreciate the detailed, constructive critiques. Below we describe our response to each of the suggestions (designated by “>”). We also indicated the changes in the text and information inserted. We hope that you will find this version to be substantially improved and ultimately acceptable for publication.
Kind Regards,
Reviewers input and our responses:
Reviewer 2: The only advice I would like to give to the authors is to mention literature data in the discussions that demonstrate how safe this tea tree oil is from both an environmental and human health point of view.
>: We thank the reviewer 1 for this suggestion, as we believe it is important to mention and will ultimately improve the quality of the manuscript. The suggested information was inserted in the text on line 279-287:
“Tea Tree Oil (TTO) was approved by the European Union (EU) and included in the positive list of the EU, in Annex I of Directive 91/414/EEC for registration of Plant Protection Products. TTO is classified as a low risk substance in Europe, for which establishment of Maximum Residue Limits (MRL) is not required. As no residue has been established, no Protected Health Information (PHI) is required. However, it depends on local aouthorities regulation (currently ranged between 0-4 days). In addition, we decided to choose TTO because it is extracted by steam distillation from renewable plants easy to grow, cost, efficacy, environmental and human consideration. This oil is in large use in cosmetics and in medicine.”
Reviewer 3 Report
The authors investigated the induction of systemic resistance in field-grown banana plants to Fusarium wilt and greenhouse-grown tomato plants to bacterial disease by tea tree oil, and evidence shown in this manuscript indicates that TTO provides protection to the plant independently of the fungicide effect and is able to prime tomato plants to have a strong defense reaction to subsequent challenges, such as mechanical wounding.
- Many plant essential oils can act as bio-fungicide and bio-bactericide. Why do you choose essential tea tree oil? The reason should be shown to readers.
- Some plant essential oil has function to activate SAR and ISR in plants, which has also been reported before. However, the long-term efficacy of TTO in field-grown banana plants is still exciting. One of the most important results in this MS is table 1 but the method is not clear. How many plant samples are tested to get disease incidence?
- In Abstract, there is no detailed TTO application results.
- Some results especially some figures and tables are not rigorous. Fig.2B, the table title should be β-1,3-glucanase. All the following should be corrected in Fig. 2: “Guaiacol.Peroxidase.”, “1,3CGlucanase..”, and all coordinates marks.
- Section 4.8 and 4.9, there are no primer sequences of analyzed genes and no detailed instrument information. Expression of the methods is very confusing.
- L557 “Relative quantification (Rq) was calculated using 2-ΔΔCq [44, 48].” and L546 “three technical repetitions for each sample”, but no repetition or error bar can be found in Fig. 3-Fig. 7.
- L534-557 does not belong to this part “RNA extraction and reverse transcription of tomato plants”.
- L225, 329 appear negative feedback, and relevant documents and explanation should be added to prove.
- L355-370, one sentence as a paragraph is not suitable. There is no secondary title, and this section in wrong position.
Author Response
Dear Editors and Reviewers:
We are very happy with the positive reception of our manuscript, and we greatly appreciate the detailed, constructive critiques. Below we describe our response to each of the suggestions (designated by “>”). We also indicated the changes in the text and information inserted. We hope that you will find this version to be substantially improved and ultimately acceptable for publication.
Kind Regards,
Reviewers input and our responses:
Reviewer 3:
- Many plant essential oils can act as bio-fungicide and bio-bactericide. Why do you choose essential tea tree oil? The reason should be shown to readers.
>: We thank reviewer 2 for the detailed and constructive suggestions. We agree that the reason of choosing TTO is important for readers. We have inserted Information regarding that in the text on the line line 279-287.
“Tea Tree Oil (TTO) was approved by the European Union (EU) and included in the positive list of the EU, in Annex I of Directive 91/414/EEC for registration of Plant Protection Products. TTO is classified as a low risk substance in Europe, for which establishment of Maximum Residue Limits (MRL) is not required. As no residue has been established, no Protected Health Information (PHI) is required. However, it depends on local aouthorities regulation (currently ranged between 0-4 days). In addition, we decided to choose TTO because it is extracted by steam distillation from renewable plants easy to grow, cost, efficacy, environmental and human consideration. This oil is in large use in cosmetics and in medicine.”
>: In addition to that, we have rephrased and highlighted our reasons to choose TTO in the lines 288-300:
“The essential tea tree oil (TTO) was shown to be have antiseptic, fungicide and bactericide properties [27, 28, 36, 29] and in the last decade has been used as an effective fungicide on numerous crops against plant pathogens [33, 37, 30, 31]. The fungicidal and antimicrobial activities of TTO against fungal and other pathogens arise from its ability to alter the permeability of membrane structures [27, 36, 38, 29]. In yeast cells and isolated mitochondria the extract components of the tea tree (M. alternifolia) destroy cellular integrity, inhibit respiration and ion transport processes, and increases membrane permeability [27, 36, 38, 29]. Previous studies showed that formulated TTO (Timorex Gold®) effectively controlled black Sigatoka in banana plantations [34, 33, 37] by exhibiting a strong curative activity based upon disruption of cell membrane and destruction of the hyphae wall of Mycosphaerella fijiensis in infected leaves [37]. Nevertheless, no reports exist in the literature showing that TTO can be an inducer of systemic resistance in plants against pathogens.”
- Some plant essential oil has function to activate SAR and ISR in plants, which has also been reported before. However, the long-term efficacy of TTO in field-grown Submission Date Date of this review banana plants is still exciting. One of the most important results in this MS is table 1 but the method is not clear. How many plant samples are tested to get disease incidence?
>: We agree that the results shown in the table are very important. However we also agree that the way it was shown was confusing. The reviewer 1 also raised some questions about the table being confusing. To solve that, we have erased the table and presented the results in the text in a more clear way.
- In Abstract, there is no detailed TTO application results.
>: In order to respect the journal formatting guidelines in regards to maximum number of words in the abstract, we have chosen the most important information to summarize our manuscript. Even though we find the TTO application results important, we have prioritized other results to highlight in the abstract, such as the SAR and ISR data. However, we believe it is very well explained and detailed in the manuscript text.
- Some results especially some figures and tables are not rigorous. Fig.2B, the table title should be β-1,3-glucanase. All the following should be corrected in Fig. 2: “Guaiacol.Peroxidase.”, “1,3CGlucanase..”, and all coordinates marks.
>: The figure has been corrected according to the reviewer's notes. Line 161.
- Section 4.8 and 4.9, there are no primer sequences of analyzed genes and no detailed instrument information. Expression of the methods is very confusing.
>: We recognize that this section could be improve. Therefore, we have added more info and inserted the list of primers and their respective nucleotide sequences in the table 2. We improve the text on the lines 502-589.
- L557 “Relative quantification (Rq) was calculated using 2-ΔΔCq [44, 48].” and L546 “three technical repetitions for each sample”, but no repetition or error bar can be found in Fig. 3-Fig.
>: Normally, gene expression levels data are transformed to log2. This transformation renders that all upregulated genes are positive and down-regulated genes negative. After the log2 transformation, only significant data are plotted in the graphs, therefore, there is no need for errors bars. Inserted line: 586-589.
- L534-557 does not belong to this part “RNA extraction and reverse transcription of
tomato plants”.
>: We agree that this section is out of place. We have corrected and detailed all information separately from banana, tomato plants and general information, such as the method used for the real time quantitative PCR. Line 531-539.
- L225, 329 appear negative feedback, and relevant documents and explanation
should be added to prove.
>: Negative feedback is part of the priming phenomena. In summary, plants activate their defenses after a first challenge, deactivate it in a negative feedback manner and after a second challenge it will show bigger and faster defense activation. In this sense, we have cited many papers stating that pattern.
- L355-370, one sentence as a paragraph is not suitable. There is no secondary title,
and this section in wrong position.
>: We agree. We have merged the section with the one following and have repositioned it. Line 370-378; 458-460.
Reviewer 4 Report
The article entitled: " Tea tree oil induces systemic resistance against Fusarium wilt in banana and Xanthomonas infection in tomato plants" is well presented and structured. All the experiments have been carried out properly, and the data analyzed and interpreted as expected. Considering these premises, I recommend the paper for publication after minor revisions.
- Figure 1 A and B: I suggest arrange the title and removing all the points
- I suggest to authors to add, in material and method section, a table with primers used to evaluate the expression of genes considered in the work.
Author Response
Dear Editors and Reviewers:
We are very happy with the positive reception of our manuscript, and we greatly appreciate the detailed, constructive critiques. Below we describe our response to each of the suggestions (designated by “>”). We also indicated the changes in the text and information inserted. We hope that you will find this version to be substantially improved and ultimately acceptable for publication.
Kind Regards,
Reviewers input and our responses:
Reviewer 4:
- Figure 1 A and B: I suggest arrange the title and removing all the points.
>: We thank the reviewer 3 for the suggestions. The title was changed and the arrows were removed from the figure as suggested, the figure ultimately looks better and easier to understand. The title was changed as follows: Line 136-141:
“Acquired systemic resistance (SAR) and induced systemic resistance (ISR) via priming is activated by TTO in banana trees infected Fusarium oxysporum f. sp. cubense (Foc) race 1. Leaf sprays of TTO on infected mother banana trees induced protection against Fusarium oxysporum f. sp. cubense (Foc) race 1 in new plants (daughters) developed.“
- I suggest to authors to add, in material and method section, a table with primers used to evaluate the expression of genes considered in the work.
>: The list of primers and their nucleotide sequences used in this manuscript was inserted in line 569-570.
Round 2
Reviewer 3 Report
It is acceptable.